# Prevalence of Reproductive Disorders including Mammary Tumors and Associated Mortality in Female Dogs

**DOI:** 10.3390/vetsci8090184

**Published:** 2021-09-04

**Authors:** Claire Beaudu-Lange, Sylvain Larrat, Emmanuel Lange, Kevin Lecoq, Frédérique Nguyen

**Affiliations:** 1Clinique Vétérinaire de la Pierre Bleue, 1 Rue de la Prairie, 35550 Pipriac, France; lange.emmanuel@gmail.com (E.L.); kevinlecoq35@gmail.com (K.L.); 2Clinique Vétérinaire Benjamin Franklin, 38 Rue du Danemark/ZA Porte Océane, 56400 Brech, France; sylvainlarrat@yahoo.fr; 3Université de Nantes, Oniris, Inserm, CRCINA, F-44000 Nantes, France; frederique.nguyen@oniris-nantes.fr

**Keywords:** dog, pyometra, spaying status, mammary tumor, radical mastectomy, survival

## Abstract

Female dogs, especially intact or neutered lately, are at increased risk for reproductive disorders including mammary tumors (MTs). This retrospective study evaluated the prevalence of reproductive pathology and associated mortality in a cohort of female dogs presented at a single veterinary clinic. The medical records of female dogs born in 2000–2003 were reviewed. The study included 599 cases, of which 293 were followed up until death. Causes of death were analyzed according to the spaying status. Among the 599 female dogs, 306 were intact (51%), 50 (8%) had been spayed before 2 years of age (ES, early spaying), and 243 (41%) after 2 years (LS, late spaying). During their lifetime, 79 dogs (13.2%) developed pyometra, and 160 (26.7%) a mammary tumor. Among the 293 dogs with complete follow-up, 103 (35.1%) had at least one MT during their lifetime, of which 53 (51.5%) died of their mammary cancer. Spayed (ES + LS) female dogs had a 4-fold decreased risk of dying from mammary cancer (OR = 0.23, 95% CI 0.11–0.47, *p* < 0.0001) compared to intact females. In this low-sterilization rate population, MTs developed in 35.1% of female dogs over their lifetime and was the cause of death in half of them.

## 1. Introduction 

Information regarding morbidity and mortality associated with female reproductive pathology among client-owned female dogs during their lifespan is scarce. Old studies are commonly cited to advise spaying before first or second oestrus, in order to prevent mammary tumors (MTs) or pyometra [1,2,3]. But in the last decade, Beauvais pointed out the low statistical reliability of these former studies [4], and many epidemiological studies have since shown that early spaying was significantly associated with increased risks of osteoarthritis, cruciate ligament rupture, immune-mediated diseases, epileptical disorders, or some cancers in female dogs, depending on the breed [5,6,7,8,9,10,11,12]. The idea subsequently emerged that early sterilization was not necessarily conclusive as a preventive measure, at least in some breeds, such as Labrador Retrievers, Golden Retrievers, Rottweilers, and Vizslas.

As early sterilization was hypothesized to prevent the development of MTs in female dogs [1,2,3], morbidity and mortality associated with reproductive disorders in female dogs have been hypothesized to be high in the context of a low sterilization rate. However in Western European countries, spaying of female dogs is not as frequent as in Anglo-Saxon countries [10], and the prevalence of MTs according to the neutering status has not been studied in such countries. 

The mortality of female dogs with MTs is strongly dependent upon the surgical procedures performed, i.e., mastectomy and ovariectomy. More than half of female dogs operated for MTs with partial mastectomy may develop a new tumor in the ipsilateral mammary line [13,14], but sterilization at the time of MT surgery significantly diminishes such recurrences [13]. In cats with mammary tumors also, it is recognized that radical mastectomy is associated with longer survival than partial mastectomy [15,16], and that bilateral radical mastectomy further increases survival [17].

The first objective of this study was to determine the frequency of reproductive disorders over the lifetime of female dogs according to their spaying status (early-, late-spayed or intact). The second objective was to analyze the causes of death of these female dogs, and notably the role of reproductive disorders. Specifically, in dogs with mammary tumors, the third objective was to determine which clinical factors affected survival.

## 2. Materials and Methods

### 2.1. Patients

The medical records from the veterinary clinic of Pipriac (France) were reviewed to gather information on female dogs born from 1 January 2000, to 31 December 2003. Cases were included if at least one complete examination was carried out after 6 years of age. An ovariectomy was systematically recommended for all female dogs before puberty (around 6 months of age). In female dogs that had undergone ovariectomy, the exact date of sterilization was recorded if available, and all cases were divided into 3 groups: intact females, early-spayed (ES, before 2 years of age), and late-spayed (LS, after 2 years of age) females.

### 2.2. Survey of Reproductive Pathology including Mammary Tumors

The occurrence of reproductive disorders (pyometra, benign and malignant mammary tumors, urinary incontinence) was recorded for each dog. 

As soon as a mammary tumor was discovered, a unilateral radical mastectomy with concurrent ovariectomy was recommended regardless of tumor size. Cancer staging was systematically carried out before surgery in case of MTs larger than 2 cm in diameter, or if the MT had grown rapidly recently or in case of ulceration. In the other cases, the clinical staging was performed post-surgically if the MT was diagnosed as malignant by histopathology. Cancer staging consisted of 3 thoracic X-rays (ventrodorsal, left lateral, and right lateral views) and ultrasound examination of inguinal and iliac lymph nodes. Mastectomy was performed only in the absence of iliac or substernal lymph node modification suggestive of nodal metastasis, and absence of detectable pulmonary metastases. A two-step surgery was performed with single anesthesia: ovariectomy first, with a short incision on the white line; the unilateral radical mastectomy was then performed with systematic excision of the ipsilateral inguinal lymph node. MTs and inguinal lymph nodes were routinely subjected to histopathological analysis. No adjuvant chemotherapy-based treatment was undertaken except for one female, diagnosed with a mammary carcinoma at 3 years of age.

### 2.3. Causes of Death

Two hundred and ninety-three (293) of the 599 female dogs (49%) could be followed up until death. The cause of death was determined according to the last clinical examination, blood tests, and medical imaging findings. Causes of death related to reproductive pathology included pyometra with peritonitis and/or renal failure, locoregionally advanced mammary cancer, and mammary cancer with distant metastases. To be included in the group of death due to MT metastases, bitches had to have X-ray confirmed lung metastases. The other causes of death were classified into the following categories: non-reproductive tract cancers (liver, spleen, urinary bladder, kidney, bone cancers, lymphoma, cutaneous mast cell tumor, others), respiratory diseases, cardiac disorders, endocrine diseases (diabetes mellitus, diabetes insipidus, Cushing syndrome), locomotor difficulties with peripheral or central nervous symptoms, azotemia, chronic hepatitis or cirrhosis, pancreatitis, accident, aggressive behavior, or miscellaneous/unknown cause.

### 2.4. Statistical Analyses 

Survival analyses were performed with R 3.6.2 and its packages survival and survminer. The effects of early spaying and MTs on survival were analyzed with log-rank tests (for categorical variables) and Cox proportional-hazards regression models (for continuous variables). The proportional hazard assumption was checked graphically with graphics of the scaled Schoenfeld residuals against time. Kaplan-Meier curves were drawn using Medcalc® statistical software (MedCalc Software Ltd., Ostend, Belgium). When required, normality was assessed with Shapiro-Wilk’s test. Student’s *t*-test was used to compare numerical variables between groups and help identify possible confounding factors. A logistic model approach was used to evaluate the relationship between the spaying status (intact, ES, and LS) and urinary incontinence. For all statistical tests, a *p*-value < 0.05 was considered significant.

## 3. Results

### 3.1. Patient Characteristics

One thousand one hundred and fifty (1050) females were born from 2000 to 2003, of which 599 (57%) met the inclusion criteria, and 293 were followed up until death. Three hundred and six animals (306/599, 51%) were lost to follow-up. The most represented breeds were the Labrador Retriever (58/599, 9.7%), Brittany Spaniel (38/599, 6.3%), Yorkshire Terrier (33/599, 5.5%), and Poodle (32/599, 5.3%) breeds. Crossbred dogs made up 20% of the cases (Table 1). The median body weight at the first physical examination after 6 years of age was 15.0 kilograms (mean ± SD, 18.0 ± 13.1 kg). The weight of the bitches was less than 10 kg in 41.6% of the cases (245/589), between 10 and 25 kg in 29.4% (173/589), and more than 25 kg in 171 bitches (29.0%) (there were 10 missing data). 

The cohort comprised 306 intact females (51.1%), 50 ES females (8.3%), and 243 LS females (40.6%). The medical reasons for late ovariectomy were pyometra (51/243 dogs, of which 3 also had ovarian cancer), pseudopregnancy (*n* = 7), unwanted pregnancy (*n* = 16), dystocia (*n* = 7), abnormal heat duration or metrorrhagia (*n* = 4), vaginal mass (*n* = 4), ovarian cancer (*n* = 2), diabetes mellitus (*n* = 1), inguinal hernia (*n* = 1) or ovariectomy at the time of mastectomy for a mammary tumor (*n* = 47/243). In the other cases, LS was performed for the owners’ convenience.

### 3.2. Reproductive Pathology including Mammary Tumors

During the survey, 79 dogs (13.2%) suffered from pyometra, 21 dogs (3.5%) had urinary incontinence, 160 (26.7%) had a mammary tumor, and 5 (0.8%) had an ovarian cancer.

Of the 79 bitches with pyometra (2 ES, 51 LS, 26 intact dogs), 51 (65%) had ovariohysterectomy, 13 (16%) were treated medically (with aglepristone and antibiotherapy) and 15 (19%) were euthanized. The mean age of female dogs with pyometra was 9.3 ± 3.4 years and was significantly younger in dogs that had ovariohysterectomy (8.6 ± 3.2 years) than in dogs medically treated or euthanized (11.0 ± 3.4 years, *p* = 0.009).

Urinary incontinence occurred in 21 dogs: 7 (14%) of the 50 ES females, 12 (4.9%) of the 243 LS females, and 2 (0.7%) of the 306 intact females. The mean age at the diagnosis of urinary incontinence was 12.7 ± 2.5 years (*n* = 20). Compared to intact females, ES females had a 25-fold higher risk of developing urinary incontinence (Odds Ratio OR 24.7, 95% CI 5.0–123.0, *p* < 0.001), and LS females had an 8-fold increased risk (OR 7.9, 95% CI 1.8–35.6, *p* = 0.008). 

One hundred and sixty (160) females (26.7% of the global population) presented a mammary tumor. MTs occurred in 2 ES females (1% of ES females), 13 LS females (5.3% of LS females), and 145 intact females (47.4% of intact females). Compared to LS and intact females, ES dogs were at a significantly lower risk of developing MTs (OR = 0.10, 95% CI 0.01–0.41, *p* < 0.001).

The mean age at the MT diagnosis was 10.3 ± 2.8 years (median 10.0) in the global population, 10.8 ±1.9 (median 11.0) in LS females, and 10.4 ± 2.8 years (median 10.0) in intact females. Of the 2 ES female dogs that developed MTs, one was 1-year-old at the MT diagnosis, was sterilized concurrently to mastectomy, and died of renal failure at 13 years of age. The second bitch developed her MT at the age of 5 years and died of lymphoma at 8 years of age.

Of the 160 female dogs with MTs, only 59 (36.9%) had surgery, while in the other 101 cases, mastectomy was declined by the owners, mainly for financial reasons, or in the case of end-stage MTs (19 cases). The type of surgery was a radical mastectomy and ovariectomy in 44 dogs (74.5%), radical mastectomy without ovariectomy in 3 dogs (5.1%), regional mastectomy plus ovariectomy in 3 dogs, regional mastectomy without ovariectomy in 3 dogs, and unknown in 6 cases operated outside of the clinic. Female dogs that underwent surgery were significantly younger (mean 8.9 ± 2.8 years, median 9.0) than those that did not (mean 11.1 ± 2.5 years, median 11.0, *p* < 0.001).

The mean MT size at the first diagnosis was 2.2 ± 3.8 cm in the 59 dogs with MTs which benefited from a mastectomy (median 0.8 cm, *n* = 40, 19 missing data) and 4.1 ± 4.8 cm (median 1.5 cm, *n* = 83, 18 missing data) in the 101 dogs that did not benefit from surgery (the difference was not significant, *p* = 0.061, Mann-Whitney U test). The MT sizes (Figure 1) did not follow a normal distribution (Shapiro-Wilk test, *p* < 0.001). 

There were no significant differences in tumor size in the 59 dogs that benefited from a mastectomy (most of which with concurrent ovariectomy, grey plots, *n* = 40 cases with known tumor size, 19 missing data) and in the 101 dogs that were not operated on (black plots, *n* = 83 cases with known tumor size, 18 missing data). MT: mammary tumor.

In the 59 dogs treated by mastectomy, 16 (27.1%) had a benign mammary tumor, 19 (32.2%) had an invasive mammary carcinoma, 3 (5.1%) had a malignant myoepithelioma, 2 (3.4%) had a mammary carcinoma in situ, 1 (1.7%) had a mammary sarcoma, but in 18 cases (30.5%), either the owners declined histopathological analysis, or the pathological report was not retrieved from the veterinary clinics in which the bitch was operated. The mean age at the MT diagnosis was 8.6 ± 2.1 years (median 8.5) in dogs with benign MTs, 9.6 ± 2.6 years (median 10.0) in dogs with malignant MTs, and 8.3 ± 3.5 years (median 8.0) in bitches with unknown histological data (not significant difference, *p* = 0.169). All radical mastectomies had clear margins.

Following mastectomy, 23 of the 59 dogs treated by mastectomy (38.9%) developed MT recurrence in the contralateral mammary chain, after a mean of 2.1± 2.4 years (median 1.0 year). Nine of them (39.1%) had a second unilateral radical mastectomy, while in the other 14 dogs, the owners declined a second mastectomy. Among the 101 bitches with MTs whose owners declined mastectomy, 68 were regularly presented for annual controls, of which 27 showed a progressive increase in MT size over an average of 2.3 ± 1.3 years (median 2.0), after which MTs suddenly grew very quickly, or ulcerated, or metastasized to the lymph nodes with or without distant metastases. 

### 3.3. Causes of Death

Causes of death are summarized in Table 2. The first cause of death was cancer (both mammary and non-mammary), which represented 34.5% of the causes of death overall (101/293), 42.4% of the causes of death in intact females (59/139), 32.3% in ES females (10/31), and 26.0% in LS females (32/123). The higher proportion of cancer-related death in intact females (59/139, 42.4%) was statistically significant compared to ES + LS dogs (42/154, 27.3%, Fischer’s exact test, *p* = 0.007). Compared to intact female dogs, spayed females (either early or late) had a 2-fold decreased risk of dying from cancer (OR = 0.51, 95% CI 0.31–0.83). 

Reproductive disorders were the cause of death in 23.2% of the dogs (68/293), including 11.4% of LS dogs (14/123), 38.8% of intact dogs (54/139), and none of the ES dogs. The leading cause of death due to reproductive pathology was mammary cancer (*n* = 53), followed by pyometra (*n* = 12), and rare ovarian carcinomas (*n* = 3). 

Among the 293 female dogs followed-up until death, 103 (35.2%) presented at least one MT throughout their lifetime, corresponding to 63/139 intact females (45.3%), 38/123 LS females (30.9%), and 2/50 (4%) of the ES females. Mammary cancer was the cause of death in 18.1% of the global cohort (53/293), corresponding to 51.5% of the 103 female dogs with MTs and complete follow-up. MT was the cause of death in 42/139 intact females (30.2%), 11/123 LS females (8.9%), but none of the ES females. Compared to intact females, LS female dogs had a 4-fold decrease in their risk of dying from mammary cancer (OR = 0.23, 95% CI 0.11–0.47, *p* < 0.0001).

Of the 59 females who operated on for MTs, 35 had complete follow-up. Ten of them (28.6%) died from their mammary cancer, after a mean time of 3.2 ± 2.7 years (median 2.5 years): 2 developed distant metastases from their primary mammary malignancy (1 with sarcoma-developed lung metastasis 3 years after mastectomy, and 1 with node-positive mammary carcinoma-developed lung metastasis 1 year after mastectomy), 1 dog died 3 weeks after regional mastectomy due to skin invasion by an inflammatory mammary carcinoma, and 7 females died from distant metastases or locally aggressive carcinoma after the onset of a contralateral recurrence that had not been removed surgically. The cause of death was unrelated to cancer in 17 cases (48.6%), non-mammary cancer in 4 cases (11.4%), and undetermined in 4 cases (11.4%). Of note, none of the dogs that had a second surgery for a local MT on the contralateral chain (*n* = 9) died of MT (3 died from renal failure, 2 from neurological disorders, 1 from liver cancer, 3 from non-cancerous diseases). 

Among the 101 bitches with MTs whose owners declined mastectomy, 33 were lost to follow-up, while 68 were followed up until death. Of them, 43 bitches (63.2%) deceased from their MT due to rapid local growth, ulceration, and/or metastasis. The cause of death was unrelated to cancer in 13 cases (19.1%), non-mammary cancer in 4 cases (5.9%), and undetermined in 8 cases (11.8%). 

Compared to female dogs with MTs treated by mastectomy (*n* = 35 with complete follow-up), those that did not benefit from a mastectomy (*n* = 68 with complete follow-up) had a 4-fold increased risk of dying from their mammary cancer (OR = 4.1, 95% CI 1.8–10.4, *p* = 0.0008). 

### 3.4. Survival Analyses

Female dogs followed up until death (*n* = 293) died at a mean age of 12.4 ± 2.7 years (median 13.0, range 5–18). Intact females died at a significantly younger age (mean 12.0 ± 2.8 years, median 12.0, range 5–18) than ES females (mean 13.2 ± 2.4 years, median 13.0, range 7–17, *p* = 0.036) and LS females (mean 12.7 ± 2.6 years, median 13.0, range 6–17, *p* = 0.023). Age at death was not significantly different between ES and LS dogs. 

Age at death significantly differed according to the cause of death: female dogs died at a younger age of non-mammary cancers (mean 11.0 ± 2.3 years, median 11.0) than those that died of reproductive disorders including mammary cancers (mean 12.5 ± 2.5 years, median 12.0, *p* = 0.0009), and those that died of non-cancerous causes (mean 12.8 ± 2.8 years, median 13.0, *p* < 0.0001).

Out of the 103 dogs with MTs and complete follow-up, survival analyses were performed in the 84 patients that were not euthanized at the diagnosis of a final-stage mammary cancer. Overall survival time significantly depended on 3 parameters by univariate analysis: age at the diagnosis, spay status, and treatment options. Age at MT diagnosis had a significant negative effect on survival, with each additional year associated with a +32% increase in mortality (Hazard Ratio HR = 1.32, 95% CI 1.14–1.52, Cox model, *p* = 0.0001). Spayed (ES + LS) females had a 3-fold decreased mortality rate compared to intact females (HR=0.28, 95% CI 0.14–0.55, *p* = 0.0001, log-rank test). The overall survival probability at 1-year post-diagnosis was 69% in intact females, compared to 95% in spayed (ES + LS) dogs (Figure 2A).

There was a significant survival improvement in the 35 dogs that underwent surgery (mastectomy + ovariectomy in most cases) compared to the 49 dogs for which owners declined surgery (HR = 0.31, 95% CI 0.16–0.62, *p* = 0.0003). The overall survival probability at 1-year post-diagnosis was 91% in operated females, compared to 74% in non-operated dogs (Figure 2B).

By multivariate survival analysis, 2 parameters were significantly associated with overall survival of female dogs with MTs (Cox model, *p* < 0.0001): age at the diagnosis, which had a significant negative effect on survival, and the spay status, with a protective effect observed in neutered female dogs (Table 3). 

## 4. Discussion

In a previously published cohort of female Beagle dogs, follow-up indicated that MTs tended to appear around 8 years of age and then increased in incidence with age [18]. From former studies, early sterilization or sterilization before the third heat was thought to prevent MTs in female dogs [1,2,3]. Sterilization was therefore recommended for all female dogs without breeding future. Beauvais questioned these dogmas in 2012 due to statistical bias found in these former studies [4]. Moreover, some authors have questioned further the preventive interest of early sterilization, because of a higher incidence of various cancers in early-spayed female dogs, at least for some breeds such as Rottweilers, Labrador Retrievers, Golden Retrievers, or Vizslas [5,6,7,8,9,12]. This retrospective study aimed at comparing the prevalence of reproductive pathology including MTs in client-owned female dogs according to their sterilization status, to analyze the causes of death, and to evaluate the impact of MTs on life expectancy.

The birth dates 2000–2003 were chosen in order to have access to the oldest accessible computer data files on the one hand and to get a chance to have a complete follow-up until death on the other hand. Unfortunately, the rhythm and number of oestrus cycles before sterilization were not available. It is thus possible that the ES group contained some females that had had 3 periods of heat before ovariectomy.

In this cohort, only 8.3% of female dogs were spayed before 2 years of age (ES dogs): as in other Western European countries, sterilization was less widespread than in Anglo-Saxon countries [10]. Urinary incontinence affected 14% of ES females, similar to previous studies [8,19,20,21,22]. Some females (40.6%) were spayed later on during their lifespan (LS group), due to unwanted pregnancy, pseudopregnancy, pyometra, or concurrently with mastectomy. Among LS and intact females with complete follow-up, 18.3% (48/262) presented a pyometra during their lifespan, a proportion similar to that described in Sweden, with 19% of female dogs having pyometra around 10 years of age [23]. Most of them were cured but 15.0% (12/79) died from it, similar to previously reported mortality rates (3–20%) [24]. Regarding the incidence of urinary incontinence or pyometra, this cohort seems similar to previously described cohorts.

In the present study, a high and unexpected percentage of female dogs, 26.7% (160/599), presented MTs, at a mean age of 10.3 ± 2.8 years. In a study of 260,000 dogs insured in Sweden, only 13% had developed MT by the age of 10, with a mean age of 8 years at the diagnosis, with predisposed breeds such as Leonberger, Irish Wolfhound, Bernese Mountain Dog, Great Dane, Staffordshire Bull Terrier, Rottweiler, Bull Terrier, Dobermann, Bouvier des Flandres, and Airedale Terrier [25], which were poorly represented in the present study. Availability of data at the time of death is the originality and strength of the present study. Among the 293 dogs with complete follow-up, 35.2% had presented a MT during their lifespan (corresponding to 45.3% of the intact females). MTs were the cause of death of 30.2% of intact females. According to Taylor, the cumulative risk of developing a MT increases from age 5 to age 13 [18]. Jitpean reported on the rate of MTs at the age of 10, and probably underestimated the percentage of females affected by MTs during their lifespan [25]. Salas reported that adult female dogs (9 to 12 years old) were most frequently involved [26]. Indeed, Vascellari reported that MTs accounted for 54% of all tumors in female dogs [27].

The present study was retrospective. The most critical bias in such studies is the selection bias. As we enrolled all of the female dogs born between 2000 and 2003 that were recorded in our database, under the condition that at least one complete physical examination was available from 6 years of age, with known sterilization status, such a bias is unlikely to have occurred. Other sources of bias such as a relationship between owner decision regarding spaying or MT surgery and specific profiles of dogs may have been present. However, the recommendations to owners regarding spaying and treatment of MTs were consistent in time and dispensed similarly by the 3 veterinarians of the clinic during the period studied. As the choice to follow recommendations or not depended mostly on the sets of beliefs and financial situation of the owners, major sources of bias seem unlikely to be present, and comparisons with internal control groups should be statistically reliable. 

In this study, urinary incontinence was inversely associated with age at the sterilization (OR = 24.7 in ES dogs, 7.9 in LS, and 1.00 in intact females), as previously reported [8,19,20,21,22,28,29]. The relative risk of developing a MT was very low in ES dogs (OR = 0.10) compared to LS and intact females, as was the risk of dying from it (OR = 0.00 in ES dogs, 0.23 in LS dogs compared to 1.00 in intact females). This corroborates the conclusions from older studies on the preventive effect of spaying on MT development [1,2,3]. Spaying before age 2 was not very early. It may therefore not be necessary to perform very early sterilizations to achieve this protective effect, and it may be possible to reduce the deleterious effects of early spaying while maintaining the preventive ones, with later sterilizations. Unfortunately, we could not further categorize the ES females into more precise sterilization age groups but it would be interesting to determine the best spaying time point in future studies.

In all dogs with MTs, a radical unilateral mastectomy with simultaneous ovariectomy was advised. Owners were less likely to accept surgery for older female dogs, of which 18.8% (19/101) were presented at the first medical consultation with end-stage tumors, which had often been noticed for a while but worsened suddenly. Undergoing a complete radical unilateral mastectomy with ovariectomy as soon as MTs were discovered significantly increased survival of the affected dogs (HR = 0.31) compared to untreated patients, knowing that MT size was not statistically different in operated and non-operated bitches. 

Some dogs with MTs that did not benefit from a mastectomy could be followed up. They presented worsening MTs that grew slowly at first, and then suddenly acquired local and distant malignant behavior within a median time of 2.3 years. The lack of histological analysis of MTs in non-operated dogs, unfortunately, induces a bias. It has been shown that older females are more prone to invasive mammary carcinomas while younger females may have mammary carcinomas in situ, associated with much better survival [30]. This would fit the hypothesis of a continuum between benign and malignant MTs in dogs [31], or a worsening behavior of malignant MTs with time [32], and argues in favor of operating canine patients with MTs as soon as possible.

In the present study, 38.9% of the operated females developed new MTs (all contralateral) in a median time of 1 year, of which 60% were malignant. By comparison, the reported local recurrence rates of canine MTs in published data are 12–58% with a median follow-up of 3 years [14,33] or 34% at 1-year post-mastectomy [34] for malignant MTs, and either 36% or 68% for benign MTs within 31.5 months, depending on whether the dog was ovariectomized or not at the time of mastectomy [13]. Among dogs that developed contralateral recurrences, those who benefited from a second radical mastectomy died from another cause than MT in 77.7% of the cases. Among dogs operated for their MTs, 28.6% died from their MTs, compared to 42% in the study by Kristiansen, in which dogs were also treated by mastectomy and ovariectomy [13,35], and 32% in the study by Tran, in which one-third of the patients received adjuvant chemotherapy [36]. The survival difference is likely related to the fact that 27.1% of the operated dogs of the present study had benign MTs. The good survival probabilities observed in the present study argue in favor of performing a complete radical mastectomy in female dogs with MTs, as recommended for female cats with MTs [15,16,17], with simultaneous ovariectomy as previously proposed [13].

In the present study, ES and LS dogs died later (median 13 years) than intact female dogs (median 12 years). Early spaying has been claimed to protect against death from infectious disease, trauma, vascular disease, degenerative disease, but to promote immune-mediated disease and non-MT neoplasia, and has been associated with longer lifespan expectancy in dogs [11,37]. In this study, none of the ES dogs died of MTs, but non-mammary cancers were the leading cause of death (in 32.3% of ES dogs). Indeed, it has been reported in the last decade that early-spayed females developed some cancers at a higher frequency and earlier than intact females (lymphoma, cardiac and splenic hemangiosarcoma, mast cell tumors, transitional cell carcinoma of the urinary bladder, osteosarcoma), especially in some breeds [6,7,8,9,11,38]. Late spaying was associated with a lesser incidence of death from neoplasia than in ES females, perhaps due to the protective effect of estrogens against non-mammary cancer development [39]. Advising such late spaying could perhaps still protect against MTs, avoid pyometra, and reduce some complications associated with early sterilization such as urinary incontinence. Further prospective studies with larger cohorts would be needed to determine the best time of spaying for female dogs.

## 5. Conclusions

In this low-sterilization rate population, MTs developed in 35% of female dogs over their lifetime and was the cause of death in half of them. The risk of developing a MT was very low in Early Spayed dogs compared to Late Spayed and intact females. Age at the diagnosis negatively affected the survival of female dogs with mammary tumors, whereas spaying was associated with improved survival. Mammary cancer was the most prevalent cancer in intact females, while none of the Early Spayed females died from mammary cancers. Compared to intact female dogs, Late Spayed female dogs had a 4-fold decrease in their risk of dying from mammary cancer, while spayed females (either early or late) had a 2-fold decreased risk of dying from cancer regardless of its nature.

Spaying female dogs, even after 2 years of age, could reduce the risk of developing urinary incontinence, reduce the risk of mortality due to reproductive disorders, and extend life expectancy.

## Figures and Tables

**Figure 1 vetsci-08-00184-f001:**
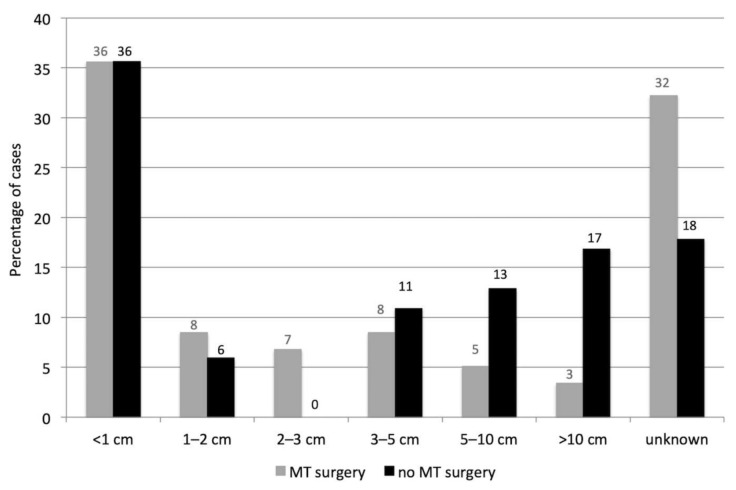
Mammary tumor size in female dogs that benefited or not from a mastectomy.

**Figure 2 vetsci-08-00184-f002:**
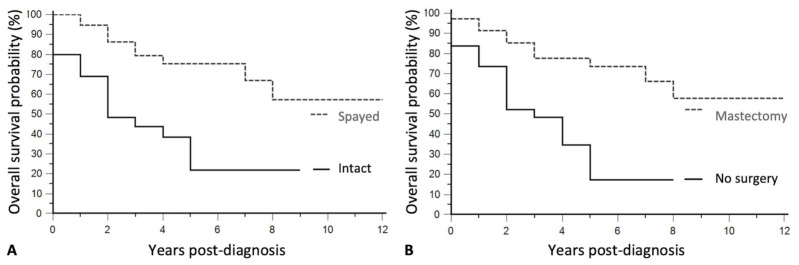
Kaplan-Meier survival curves of 84 female dogs with MTs according to the spaying status (**A**) and treatment modality (**B**). (**A**) The survival probabilities of the 39 spayed dogs with MTs (86% at 2 years post-diagnosis, grey line), were significantly better than those of the 45 intact female dogs (black line, 48% survival probability at 2 years post-diagnosis, HR = 0.28, 95% CI 0.14–0.55, *p* = 0.0001). (**B**) The survival probabilities of the 35 dogs with MTs that benefited from surgery (85% survival at 2 years post-diagnosis, grey line) were significantly better than those of the 49 dogs in which mastectomy was not performed (black line, 52% survival at 2 years post-diagnosis, HR = 0.31, 95% CI 0.16–0.62, *p* = 0.0003).

**Table 1 vetsci-08-00184-t001:** Breed distribution of the studied cohort.

Breed	Total Cohort (*n* = 599)	Intact Females (*n* = 306)	ES Females (*n* = 50)	LS Females (*n* = 243)
American Staffordshire Terrier	6 (1.0%)	2 (0.6%)	1 (0.5%)	3 (1.2%)
Beagle	9 (1.5%)	5 (1.6%)	0	4 (1.6%)
Beauceron	12 (2.0%)	6 (2.0%)	3 (1.5%)	3 (1.2%)
Belgian Shepherd	16 (2.7%)	6 (2.0%)	0	10 (4.1%)
Bernese Mountain Dog	3 (0.5%)	2 (0.6%)	0	1 (0.4%)
Bichon	24 (4.0%)	13 (4.2%)	2 (1.0%)	9 (3.7%)
Border Collie	12 (2.0%)	5 (1.6%)	0	7 (2.9%)
Boxer	16 (2.7%)	3 (1.0%)	3 (1.5%)	10 (4.1%)
Brittany fawn dog	5 (0.8%)	3 (1.0%)	0	2 (0.8%)
Brittany Spaniel	38 (6.3%)	23 (7.5%)	1 (0.5%)	14 (5.8%)
Cavalier King Charles	11 (1.8%)	3 (1.0%)	0	8 (3.3%)
Cocker spaniel	14 (2.3%)	11 (3.6%)	1 (0.5%)	2 (0.8%)
Collie	8 (1.3%)	3 (1.0%)	1 (0.5%)	4 (1.6%)
Continental Toy Spaniel	7 (1.2%)	3 (1.0%)	0	4 (1.6%)
Coton de Tuléar	5 (0.8%)	4 (1.3%)	0	1 (0.4%)
Cross-bred dogs	118 (19.7%)	55 (18.0%)	18 (9.0%)	45 (18.5%)
Dachshund	26 (4.3%)	16 (5.2%)	0	10 (4.1%)
German Shepherd	15 (2.5%)	4 (1.3%)	1 (0.5%)	10 (4.1%)
German Shorthaired Pointer	4 (0.7%)	3 (1.0%)	0	1 (0.4%)
Golden Retriever	16 (2.7%)	6 (2.0%)	0	10 (4.1%)
Labrador Retriever	58 (9.7%)	27 (8.8%)	10 (5.0%)	21 (8.6%)
Lhassa Apso	6 (1.0%)	3 (1.0%)	0	3 (1.2%)
Newfoundland dog	4 (0.7%)	4 (1.3%)	0	0
Pekingese dog	3 (0.5%)	1 (0.3%)	0	2 (0.8%)
Pinscher	10 (1.7%)	2 (0.6%)	3 (1.5%)	5 (2.1%)
Poodle	32 (5.3%)	18 (5.9%)	3 (1.5%)	11 (4.5%)
Pyrenean Mountain dog	3 (0.5%)	3 (1.0%)	0	0
Rottweiler	21 (3.5%)	11 (3.6%)	0	10 (4.1%)
Setter	17 (2.8%)	14 (4.6%)	0	3 (1.2%)
Shih Tzu	12 (2.0%)	6 (2.0%)	1 (0.5%)	5 (2.1%)
St. Bernard	3 (0.5%)	0	0	3 (1.2%)
West Highland White Terrier	3 (0.5%)	2 (0.6%)	0	1 (0.4%)
Yorkshire Terrier	33 (5.5%)	22 (7.2%)	1 (0.5%)	10 (4.1%)
Other breeds (*)	29 (4.8%)	17 (5.6%)	1 (0.5%)	11 (4.5%)
Total	599 (100%)			

(*) Dalmatian, Doberman, Dogue de Bordeaux, Fox Terrier, Jack Russell Terrier, Korthals Griffon, Pyrenean Shepherd, Siberian Husky, Weimaraner: 2 dogs (0.3%) each; Airedale Terrier, American bulldog, Basset hound, Chihuahua, German Wirehaired Pointer, French Bulldog, Pug, Scottish Terrier, Springer Spaniel, Tibetan Spaniel, Whippet: 1 dog (0.2%) each. ES: early-spayed females (before 2 years of age). LS: late-spayed females (after 2 years of age).

**Table 2 vetsci-08-00184-t002:** Causes of death of female dogs followed-up until death (N s = 293) according to their sterilization status.

Causes of Death		ES Group (*n* = 31)	LS Group (*n* = 123)	Intact Eemales (*n* = 139)
**Non-Mammary Cancer (*n* = 48)**		**10 (32.3%)**	**21 (17.1%)**	**17 (12.2%)**
	Lymphoma (*n* = 7)	2 (6.5%)	2 (1.6%)	3 (2.2%)
	Liver cancer (*n* = 5)	2 (6.5%)	2 (1.6%)	1 (0.7%)
	Oral cancer (*n* = 5)	1 (3.2%)	3 (2.4%)	1 (0.7%)
	Spleen malignancy (*n* = 4)	2 (6.5%)	2 (1.6%)	0
	Hemangiosarcoma (*n* = 3)	0	2 (1.6%)	1 (0.7%)
	Lung cancer (*n* = 3)	2 (6.5%)	1 (0.8%)	0
	Soft tissue sarcoma (*n* = 3)	0	3 (2.4%)	0
	Pancreatic cancer (*n* = 2)	0	1 (0.8%)	1 (0.7%)
	Osteosarcoma (*n* = 2)	0	2 (1.6%)	0
	Other sites (*n* = 14) ^1^	1 (3.2%)	3 (2.4%)	10 (7.2%)
**Reproductive pathology (*n* = 68)**		**0**	**14 (11.4%)**	**54 (38.8%)**
	Mammary cancer (*n* = 53)	0	11 (8.9%)	42 (30.2%)
	Pyometra (*n* = 12)	0	2 (1.6%)	10 (7.2%)
	Ovarian carcinoma (*n* = 3)	0	1 (0.8%)	2 (1.4%)
**Other causes (*n* = 177)**		**21 (67.7%)**	**88 (71.5%)**	**68 (48.9%)**
	Locomotor disability (*n* = 46) ^2^	6 (19.4%)	22 (17.9%)	18 (12.9%)
	Renal failure (*n* = 20)	2 (6.5%)	11 (8.9%)	7 (5.0%)
	Heart failure (*n* = 15)	1 (3.3%)	12 (9.8%)	2 (1.4%)
	Traffic accident (*n* = 11)	1 (3.3%)	4 (3.3%)	6 (4.3%)
	Hepatic disorders (*n* = 9) ^3^	1 (3.3%)	5 (4.1%)	3 (2.2%)
	Endocrine diseases (*n* = 8) ^4^	2 (6.5%)	2 (1.6%)	4 (2.9%)
	Respiratory failure (*n* = 5)	1 (3.3%)	2 (1.6%)	2 (1.4%)
	Pancreatitis (*n* = 5)	0	4 (3.3%)	1 (0.7%)
	Glaucoma (*n* = 3)	1 (3.3%)	2 (1.6%)	0
	Aggressiveness (*n* = 3)	0	0	3 (2.2%)
	Miscellaneous ^5^ (*n* = 52)	6 (19.4%)	24 (19.5%)	22 (15.8%)

^1^ Other cancer sites: ocular (1), nasal/sinusal (1), urinary bladder (1), heart base (2), anal sac gland carcinoma (2), cutaneous mast cell tumor (1), generalized cancer of unknown primary (6). ^2^ Locomotor disabilities included severe osteoarthritis or suspected peripheral or central nervous system lesion with locomotor impairment. ^3^ Hepatic disorders included severe hepatitis, liver failure, and cirrhosis, excluding liver cancer. ^4^ Endocrine causes of death were diabetes mellitus, hyperadrenocorticism, and diabetes insipidus. ^5^ Miscellaneous: 38 females were euthanized or returned for cremation without notations of their cause of death; 10 bitches were euthanized due to severe symptoms that the owner did not wish to explore further (anorexia and cachexia (*n* = 5) and (1 of each) blood-vomiting, polyuro-polydipsia, coma, anemia, and pancytopenia. 3 cases were euthanized respectively for discoid lupus erythematosus, chronic otitis, digestive occlusion by a foreign body, and 1 dog died during anesthesia for surgical removal of a sternal mass of unknown nature.

**Table 3 vetsci-08-00184-t003:** Multivariate analysis of factors associated with overall survival in 84 dogs with MTs.

Variables	HR	95% CI	*p*
Age at the diagnosis (years)	1.26	1.10–1.46	0.0014
Neutered females versus intact	0.33	0.15–0.71	0.0049

## Data Availability

The data presented in this study are available on request from the corresponding author.

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
