# Peer review of "Prevalence of Reproductive Disorders including Mammary Tumors and Associated Mortality in Female Dogs"

_vetsci, 2021, doi:10.3390/vetsci8090184_

Round 1

Reviewer 1 Report

In this manuscript, the authors look retrospectively at female dogs categorizing them as early spayed (< 2yrs), late spayed (> 2yrs), and intact and determined what percentage went on to develop mammary tumors. Furthermore, of those that developed mammary tumors, they determined what percentage sought aggressive treatment and what percentage died from the tumors. The debate about when and if to spay female dogs is of high interest to general practicing veterinarians. While a similar study with many more dogs was performed in Sweden, the current study highlights the fact that they had data they could follow until demise of the animal and also had different breeds represented. This also may suggest regional differences, which could be interesting to observe.

One interesting aspect to consider in this study is to determine what female dogs were used for breeding and had puppies and if those female intact dogs had a higher or lower incidence of mammary tumors compared to those intact females that were never pregnant.

The conclusions support the data in the paper and I feel the authors did a good job of acknowledging the limitations of their study and citing the appropriate references. Overall, I have no additional comments or suggestions.

Author Response

"One interesting aspect to consider in this study is to determine what female dogs were used for breeding and had puppies and if those female intact dogs had a higher or lower incidence of mammary tumors compared to those intact females that were never pregnant".

We agree with Reviewer 1 that studying the link between parity and mammary tumor incidence in intact female dogs is of particular interest, especially because age at first birth and breastfeeding numbers have been linked with breast cancer incidence in women. Unfortunately, the numbers of litters of intact female dogs was not systematically reported by the owners and/or not systematically recorded in medical files. We believe that this interesting question cannot be addressed properly with the data available, as the high proportion of missing data would introduce too much bias.

The conclusions support the data in the paper and I feel the authors did a good job of acknowledging the limitations of their study and citing the appropriate references. Overall, I have no additional comments or suggestions.

We are very grateful to Reviewer 1 for his/her appreciation of the present work.

Reviewer 2 Report

This paper is well written and easy to read. Interesting for early sterilization.

In the last decade, Beauvais pointed out the low statistical reliability of these former studies, and many epidemiological studies have since shown that early spaying was significantly associated with increased risks of osteoarthritis, cruciate ligament rupture, immune mediated diseases, epileptical disorders or some cancers in female dogs, depending on the breed.

This is the first study that explain in clear way  the correlection between spayed dog and reproductive disorders.

The first objective of this study was to determine the frequency of reproductive disorders over the lifetime of female dogs according to their spaying status (earlylate spayed or intact).

The second objective was to analyze the causes of death of these female dogs, and notably the role of reproductive disorders. Specifically in dogs with mammary tumors.

The third objective was to determine which clinical factors affected survival.

The conclusions are very interesting for the early future

Author Response

"This paper is well written and easy to read. Interesting for early sterilization. […] This is the first study that explains in clear way the correlation between spayed dog and reproductive disorders. […] The conclusions are very interesting for the early future."

We are very grateful to Reviewer 2 for his/her appreciation of this manuscript.

Reviewer 3 Report

The manuscript "Prevalence of reproductive disorders including mammary tumors and associated mortality in 599 female dogs" by Claire Beaudu-Lange et al. show MTs developed in 35.1% of female dogs over their lifetime, and was the cause of death in half of them.

However, I have some concerns and I suggest to accept the manuscript after minor revision.

  1. The authors should change the title. Is there any way the number of dogs (599) could be changed?
  2. Fig 1 shows Shapiro wilk test, P<0.001. Where is the error bars? Why anova test not shown? Is there any reason.
  3. Please change the color of the each bars in Fig1.
  4. I like the Fig2 and Fig3. It would be great to change the color and put Mts according to treatment modality and spaying status side by side.

Author Response

I have some concerns and I suggest to accept the manuscript after minor revision.

  1. The authors should change the title. Is there any way the number of dogs (599) could be changed?

The number of female dogs (599) corresponds to the number of female dogs that were presented at the veterinary clinic of Pipriac between January 1, 2000 and December 31, 2003, and were at least aged 6 years at presentation, so the number of dogs cannot be changed. However, we have followed Reviewer 3’s advice to change the title, and have removed the number of dogs from the title in the revised manuscript.

"Fig 1 shows Shapiro wilk test, P<0.001. Where is the error bars? Why anova test not shown? Is there any reason."

Actually, Fig. 1 is a simple histogram of mammary tumor size distribution in dogs that benefited from mastectomy and dogs that did not have mastectomy; thus, error bars cannot be applied here.

The Shapiro-Wilk test was used to determine if MT size was normally distributed, which was not the case (P<0.001: the null hypothesis “the population is normally distributed” is rejected).

As MT size did not follow a normal distribution, the ANOVA test was not applicable. In the revised manuscript, we have added the name of the statistical test (Mann-Whitney U test) that indicated that mammary tumor size did not significantly differ in dogs that had mastectomy and dogs that did not have surgery for their mammary tumors.

"Please change the color of the each bars in Fig1."

Following Reviewer 3’s suggestion, we have changed the colors in Fig. 1 in the revised manuscript. We have kept black and grey tones, which are easily distinguished on black-and-white prints, but the grey tone has been darkened. For a better readability, we have added the percentages of cases over each bar.

"I like the Fig2 and Fig3. It would be great to change the color and put Mts according to treatment modality and spaying status side by side."

In order to put Fig. 2 and Fig. 3 side by side, we have renumbered them Fig. 2A and Fig. 2B in the revised manuscript; we agree with Reviewer 3 that this allows for an easier comparison of the respective effects of treatment modality and spaying status on overall survival of female dogs with MTs. Regarding colors, we have kept grey and black tones for their good readability, but we have matched the color of each legend with the color of each curve in the revised Figure 2.

We thank Reviewer 3 very much for his/her suggestions for improvements of this manuscript.

Reviewer 4 Report

the work "Prevalence of reproductive disorders including mammary tumors and associated mortality in 599 female dogs " that I reviewed it was a scientifcally sounded description of a topic, that despite lack of novelty, was quite interesting and well written.

I think that the paper is overall well written, the passages are clearly described and therefore it might be suitable for pubblication in this journal after minor corrections.

remarks:

-Abstract: remove "background", "material and methods", "results", and "conclusions"

-line 48 remove "chain" use "line"

-line 71 remove "whatever" use "regardless"

-line 334 VASCELLARI (an author) is misswritten

table 2: I would add a 5th foot note to describe what "miscellaneous" include, just for clarity

Author Response

"I think that the paper is overall well written, the passages are clearly described and therefore it might be suitable for publication in this journal after minor corrections:

Abstract: remove "background", "material and methods", "results", and "conclusions"".

Thank you for the suggestion, this has been done in the revised manuscript.

"line 48 remove "chain" use "line""

Thank you for the suggestion, this has been done in the revised manuscript.

"line 71 remove "whatever" use "regardless""

Thank you for the suggestion, this has been done in the revised manuscript.

"line 334 VASCELLARI (an author) is misswritten"

Thank you very much; the spelling has been corrected in the revised manuscript.

"Table 2: I would add a 5th foot note to describe what "miscellaneous" include, just for clarity."

Thank you for the suggestion; a 5th footnote has been added at the end of Table 2 in the revised manuscript.

We are very grateful to Reviewer 4 for his/her valuable advice.